**MINIREVIEW**
Applied and Environmental Science

# Computational Analysis of Microbial Flow Cytometry Data

Peter Rubbens,[a] Ruben Props[b]

[a]Flanders Marine Institute (VLIZ), Ostend, Belgium
[b]Center for Microbial Ecology & Technology (CMET), Faculty of Bioscience Engineering, Ghent University, Ghent, Belgium

**ABSTRACT** Flow cytometry is an important technology for the study of microbial communities. It grants the ability to rapidly generate phenotypic single-cell data that are both quantitative, multivariate and of high temporal resolution. The complexity and amount of data necessitate an objective and streamlined data processing workflow that extends beyond commercial instrument software. No full overview of the necessary steps regarding the computational analysis of microbial flow cytometry data currently exists. In this review, we provide an overview of the full data analysis pipeline, ranging from measurement to data interpretation, tailored toward studies in microbial ecology. At every step, we highlight computational methods that are potentially useful, for which we provide a short nontechnical description. We place this overview in the context of a number of open challenges to the field and offer further motivation for the use of standardized flow cytometry in microbial ecology research.

**KEYWORDS** bioinformatics, cytometry, fingerprinting, data analysis, microbial ecology, single cell, multivariate statistics

Flow cytometry (FCM) is a single-cell technology that provides an optical description of individual particles based on scatter and fluorescence information. Microbial FCM has a long history, and its first applications in the field date back to the late 1970s to investigate the physiological properties of individual cultures (1, 2). The most prevalent application in microbiology remains the quantification of cell population densities in a wide range of matrices, ranging from lab cultures to marine, freshwater, soil, and fecal samples (3–8). FCM has been applied to many types of microorganisms, mostly phytoplankton and bacteria, but other types of microorganisms include single- and multicellular fungi (9, 10) and viruses (11). For many groups of microorganisms, it has proven to be both accurate and reproducible and can generate results faster than existing plate count and marker gene approaches, such as 16S rRNA gene amplicon sequencing (6, 12). The development of online and real-time FCM facilitates the quantification of microbial community dynamics at a very high temporal resolution (13–15).

A large body of research exists on extracting biological information, in addition to cell enumeration measurements, from the multivariate single-cell data acquired by FCM. Phenotypic properties, such as size, shape, morphology, activity, membrane permeability, pigmentation, and nucleic acid content are measured in various degrees, depending on the applied cell-labeling technique (16, 17). The major ongoing wet-lab FCM developments for microbiology research can be broadly classified into (i) development and standardization of novel staining methods (17, 18) and (ii) novel laboratory protocols to efficiently extract cells from complex matrices (4, 8). Much less attention is given to computational methods that can assist in the analysis of microbial cytometry data. As a result, many microbiologists perform manual interventions during their data analysis, including decisions with respect to denoising, quality control, cell population identification and statistical analyses, on a sample-by-sample or batch-by-batch basis. This inevitably results in user biases, such as reduced reproducibility, but it can also obscure meaningful biological information not apparent from the user's own interpretation. Many

Address correspondence to Peter Rubbens, peter.rubbens@vliz.be, or Ruben Props, ruben.props@ugent.be.

Computational Analysis of Microbial Flow Cytometry Data: a step-by-step minireview highlighting the full data analysis pipeline of microbial flow cytometry data.

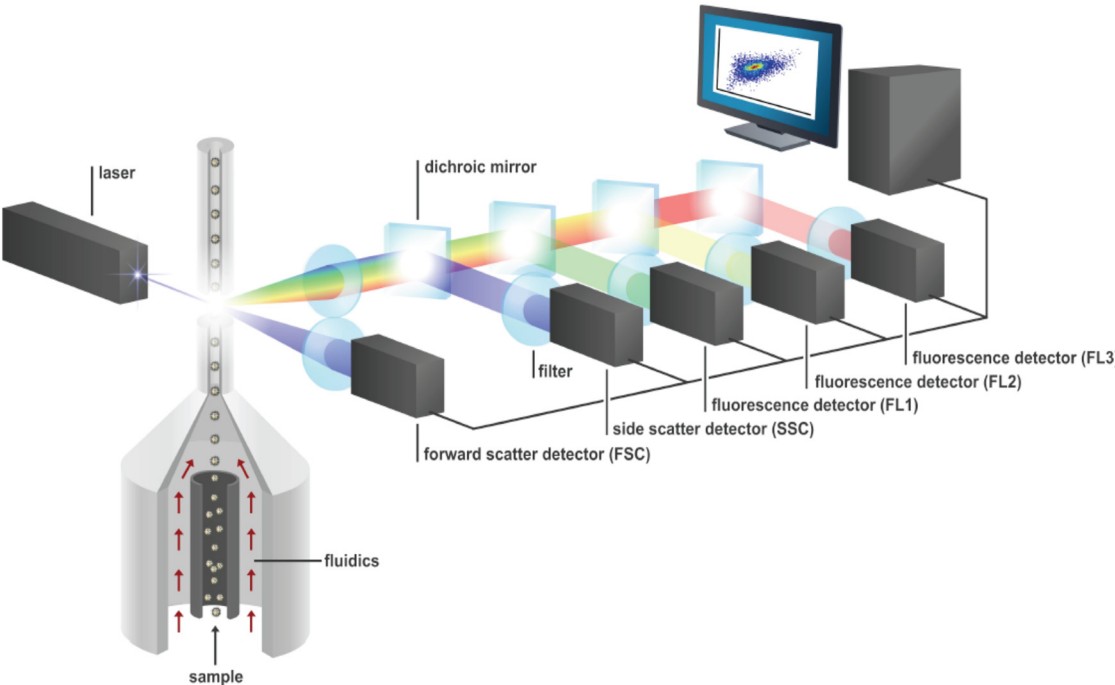

**FIG 1** Schematic overview of a flow cytometry analysis. Suspended particles are aligned one by one by hydrodynamic focusing. Next, each particle is interrogated by one or more lasers in the flow cell. The resulting scatter (FSC and SSC) and fluorescence signals (denoted "FL") of each cell are captured by multiple detectors. Fluorescence is measured at multiple wavelength intervals (three in this illustration). The electronic signals originating from these detectors are then finally transformed into digital ones.

computational methods have emerged in the biomedical research field over the past few years to address these shortcomings, grouped together under the names "FCM bioinformatics" or "computational FCM" (19, 20). These aim to facilitate and improve the objectivity, speed, and reproducibility of the data analysis. Likewise, microbiologists have the possibility to set up a dedicated data analysis pipeline to benefit from the same advantages as immunologists do.

In this review, we aim to provide a streamlined overview of the data analysis possibilities along a typical computational workflow for microbial FCM data. A demonstration of such a workflow in R can be found online at https://rprops.github.io/MSys_FCMreview/Demo.html. We highlight a number of interesting applications in which FCM is used to perform ecological studies. Additionally, we try to point to a number of challenges in the field of microbial FCM that motivate the use and development of standardized FCM for microbiology research.

## MICROBIAL FLOW CYTOMETRY

A basic overview of a flow cytometry analysis is given in Fig. 1. Suspended particles are first aligned on a one-by-one basis by means of hydrodynamic focusing. Each particle is then interrogated by one or more lasers. Optical filters allow one to measure emitted fluorescence at multiple wavelengths, next to forward scatter (FSC) and side scatter (SSC) signals. Photomultiplier tubes are used to convert the fluorescence and scatter signals to an electronic signal. The morphology of the cell is reflected in the FSC (size and shape) and SSC (intracellular complexity). The measured fluorescence is the result of autofluorescent properties (such as pigments) or the interaction with a fluorescent dye. Mostly, generic stains that target properties related to nucleic acid content, membrane integrity and other physiological aspects, such as lipid content, enzyme activity, and translational activity, are used (3, 17). The technology is fast in the sense that it is able to measure more than thousands of particles per second. It is quantitative, because each particle is described by a numeric multivariate measurement that represents a unique optical signature for each particle.

The main applications of FCM are the biological and clinical study of mammalian cells (21), also known as immunophenotyping FCM. By now, immunophenotyping data routinely represent large antibody panels of up to 28 individual biomarkers, represented by 28 different fluorescence parameters (22). These applications form the main drivers of instrument development and research. Microbial FCM has a number of different characteristics and challenges compared to immunophenotyping FCM. First, most prokaryotic cells are much smaller in size and volume than human or mammalian cells. Therefore, measurements can lie close to the detection limit of an instrument. Second, while most cells are small, the size range within which microbial cells occur is larger than for mammalian cells, covering a range between 0.2 and 500 $\mu$m. Third, microbial communities comprise high levels of phenotypic and phylogenetic complexity (e.g., 1,000s of taxa) and heterogeneity (16). As such, contrasting results concerning the establishment of multicolor staining panels for microbial communities have been reported. Single- and double-staining methods are routinely used (23, 24), with the majority of research relying on one or two general markers with phenotypic (e.g., nucleic acids or membrane permeability) (25) or phylogenetic (e.g., see reference 26) specificity. It appears much more difficult to standardize and broadly apply a triple-staining protocol, as the efficiency and stability of cell staining protocols are dependent on the bacterial taxa on which they are applied. Although successful approaches are reported in the literature (27, 28), issues such as fluorescence instability hamper their widespread use and further development (23). Therefore, microbial FCM data are characterized by data with fewer dimensions compared to immunophenotyping FCM.

## DATA ANALYSIS

A typical FCM data analysis pipeline can be broadly divided into multiple categories, of which an overview is given in Fig. 2. These include preprocessing of the data, visualization, cell enumeration of specific populations or the whole community, cytometric fingerprinting, community-level analysis and data format and storage. While some steps are necessary, others are optional and depend on the research question and experimental setup. We have summarized and ordered the computational methods that we discuss in this minireview (see Table 1). Here, we focus on software packages that are publicly available in the R statistical programming language. Note that a number of packages are also available in other languages, such as Python or Matlab.

## DATA FORMAT

FCM data are stored in flow cytometry standard (FCS) format from commercial software. The most recent version, FCS 3.1, was introduced in 2010 (29). The area (A), the height (H), and sometimes also the width (W) of fluorescence and scatter pulses are recorded for each individual particle. In addition, the file format also allows one to store metadata describing the experimental settings.

## PREPROCESSING

Before FCM data can be analyzed, a number of preprocessing steps need to be performed. Some of them are optional; others are recommended or necessary. The basic steps are available within the generic FCM software package flowCore (30).

**Compensation.** If a detector has an optical filter that captures signals coming from multiple stains, false-positive cells can be detected. Data compensation intends to correct for emission signal spillover from one stain (e.g., Syto59) into the channel designated for another stain (e.g., propidium iodide). Currently, this is only rarely applied due to the limited availability of multicolor FCM protocols to analyze microbial communities, although a few examples can be found in the literature (23, 31–34). Functions to perform compensation are incorporated into the flowCore package.

**Transformation.** Fluorescence and scatter values registered for microbial cells can differ by orders of magnitude and need to be transformed to enable separation of instrument noise and cell signals. These values exhibit linear behavior at small scales,

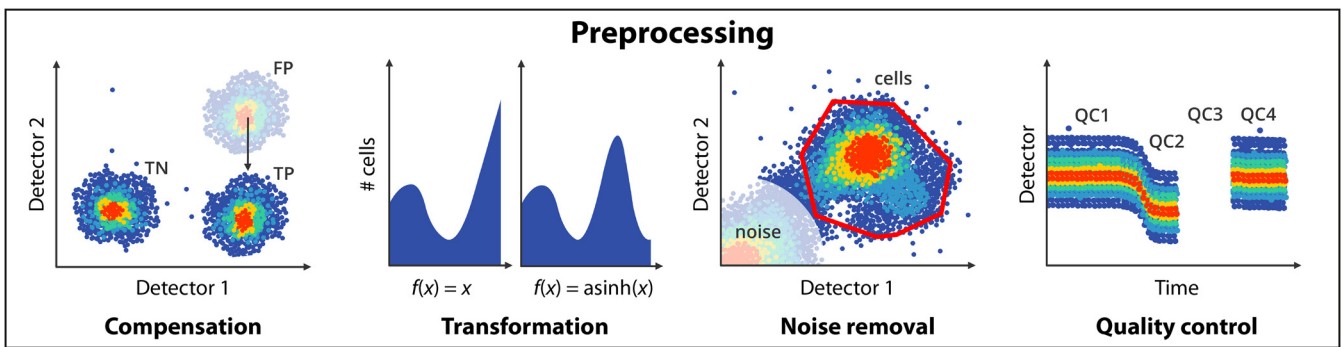

## Preprocessing

**Compensation** | **Transformation** | **Noise removal** | **Quality control**

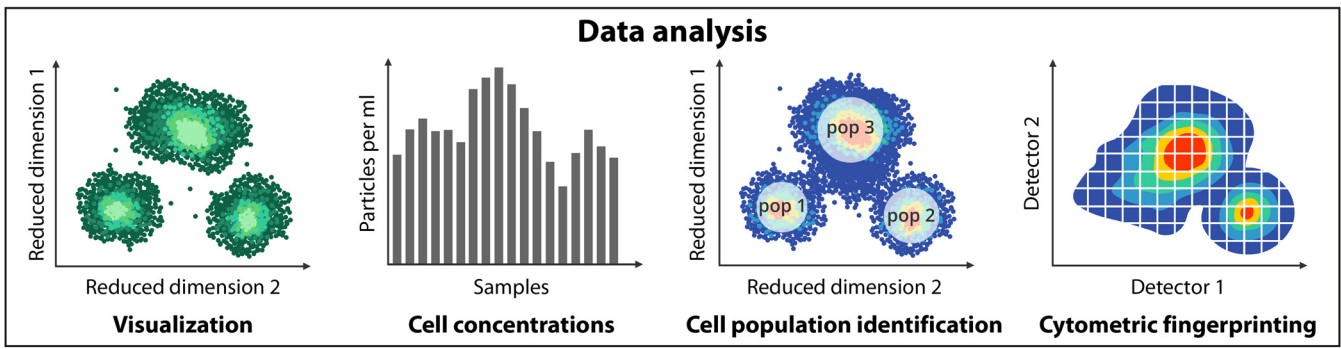

## Data analysis

**Visualization** | **Cell concentrations** | **Cell population identification** | **Cytometric fingerprinting**

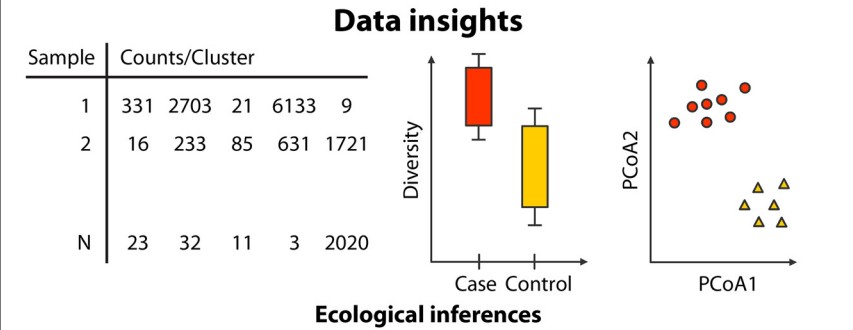

## Data insights

| Sample | Counts/Cluster | | | | |
|--------|------|------|----|------|------|
| 1 | 331 | 2703 | 21 | 6133 | 9 |
| 2 | 16 | 233 | 85 | 631 | 1721 |
| N | 23 | 32 | 11 | 3 | 2020 |

**Ecological inferences**

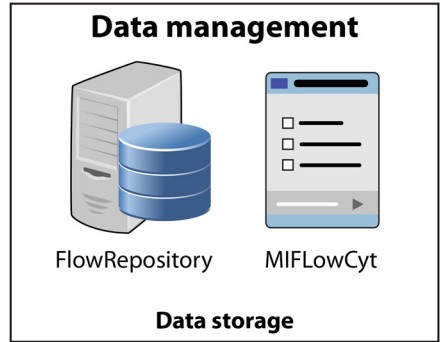

## Data management

**FlowRepository** | **MIFLowCyt**

**Data storage**

**FIG 2** Overview of a reproducible microbial FCM data analysis pipeline. First, the data are preprocessed in several steps (compensation, transformation, noise removal, and quality control). Second, the data are analyzed, which can be done in multiple ways: through visualization, determination of cell concentration, cell population identification, and/or cytometric fingerprinting. Once processed, the data can be analyzed at the community level to make ecological inferences. Upon conclusion of the analysis, the raw data are properly annotated and stored in a publicly accessible database. FP, false positives; TN, true negatives; TP, true positives; QC1 to -4, quality control measurements, respectively; pop, population; PCoA 1 and -2, principal coordinate axis 1 and 2, respectively.

but as they increase, their values increase exponentially, resulting in values that are orders of magnitude larger. Traditionally, microbial FCM data are transformed using a logarithmic function. However, measurements can be negative as well, making the logarithm unsuitable to transform the data in this case. More advanced transformations are recommended, such as the arcsine hyperbolic function or a generalized extension with one or more adjustable parameters, often referred to as the "biexponential" or "Logicle" transformation (35, 36). These transformations are available within the flowCore package. Forward and side scatter information can also be analyzed on a linear scale for heterogeneous cell populations of larger cells (37), but most microbiological applications require a transformation of the scatter parameters as well.

**Noise removal.** Instrument noise is always present in the data caused by the measurement of (in)organic particles, cell aggregates and electronic noise. Due to the small cell sizes of microbes, live cells can have fluorescence and scatter properties in a manner similar to the instrument noise. Therefore, noise removal is often performed

**TABLE 1** Overview of peer-reviewed computational methods for performing data preprocessing, visualization, cell population identification, cytometric fingerprinting and data storage

| Category | Method (reference) | Short description | Reference(s) (applied to microbial data?)[a] |
|---|---|---|---|
| Preprocessing | flowCore (30) | Basic data manipulation, gating, compensation, and transformation | 12, 72–74 |
| Preprocessing | flowTrans (37) | Optimized individual channel transformations | |
| Preprocessing | flowAI (42) | Automated denoising | 75, 76 |
| Preprocessing | flowClean (41) | Automated denoising | |
| Preprocessing | flowStats (77) | Per-channel normalization | 78 |
| Visualization | flowViz (38) | Customized FCM data visualization | 9, 79, 80 |
| Visualization | ggcyto (39) | Customized FCM data visualization with ggplot-like functionality | 74 |
| Visualization | viSNE (81) | Dimensionality reduction and visualization using t-SNE | 59, 82 |
| Visualization | UMAP (83) | Dimensionality reduction using UMAP | |
| Cell population identification | flowClust (84) | t-distribution mixture model with Box-Cox transformation | 85 |
| Cell population identification | flowEMMI (54) | Gaussian mixture model with the Bayesian information criterion | 54 |
| Cell population identification | flowPeaks (86) | k-means clustering followed by peak search and merging using a Gaussian mixture model | 87, 88 |
| Cell population identification | flowDensity (89) | Density-based sequential bivariate gating | 54, 90 |
| Cell population identification | FlowSOM (52) | Self-organizing map and meta-clustering | |
| Cell population identification | PhenoGraph (53) | k-nearest neighbor weighted graph and Louvain method for community detection | 82 |
| Cytometric fingerprinting | CHIC (61) | Two-channel histogram image comparison | 61, 72, 91 |
| Cytometric fingerprinting | flowCyBar (60) | Manual annotation of interesting regions | 60, 91–93 |
| Cytometric fingerprinting | flowDiv (62) | Fixed-binning grid over multiple two-channel combinations | 62 |
| Cytometric fingerprinting | flowFP (64) | Distribution-dependent binning in hyper-rectangles | 91, 94–96 |
| Cytometric fingerprinting | Phenoflow (12) | Fixed-binning grid and kernel density estimation over multiple two-channel combinations | 12, 76, 97, 98 |
| Cytometric fingerprinting | PhenoGMM (67) | Overclustering using a Gaussian mixture model | 67, 99 |
| Data storage | FlowRepository (70) | Public database to store and annotate FCM data | 34, 75, 100, 101 |

[a]This column highlights references in which the method has been applied to microbial FCM data.

manually, by defining rectangle, quadrant, ellipsoid, or generic polygon regions (i.e., "gates") containing the cell signals. As stated in the introduction, this can be laborious in time and introduce subjective biases (20). At the individual-sample level, optimal gates can often differ due to intersample variation. It is advised to use, whenever possible, the same gating template within a single experiment, although samples analyzed with different gating templates can still be compared using proper control samples. Negative-control samples are then necessary, which can include heat-killed samples and 0.2-$\mu$m-filtered samples, either stained and/or unstained. A gating template can be defined using the flowCore package. Denoising is usually guided by user-dependent iterative visualization of the scatterplots. FCM data visualization is supported by the flowViz and ggcyto packages (38, 39). The number of gating steps are largely dependent on the complexity of the analyzed sample. High degrees of noise may require additional signal filtering from multiple fluorescence (e.g., autofluorescence on violet laser) and/or scatter channels. In most measurements, cell aggregates, such as doublets, triplets, or chains, are measured as well and can be identified through visualizing the area and height parameter of the primary fluorescence or scatter channel. However, for microbiological applications, it remains difficult in practice, and there is no consensus yet on how to best handle cell aggregate signals. We recommend the optimization of sample preparation protocols to reduce the percentage of cell aggregates; these can include the use of filtration, ultrasonication, surfactants (e.g., Tween, Triton X-100), complexing agents (e.g., EDTA, sodium pyrophosphate), and/or $Ca^{2+}/Mg^{2+}$-free buffers (4, 40).

**Quality control.** The quality of the data and its acquisition are subject to both instrument and biological variation. The ideal data acquisition consists of the measurement of homogeneous and stable cell signal distribution during sample analysis. However, deviations can occur, for example, due to large particles that clog the system or air bubbles that cause gaps in the data. This results in aberrations in the data, such as spikes, gaps or gradual degradation of the mean fluorescence intensity. These need to be addressed and, depending on the research question, removed. While these

actions can be done manually, algorithms to detect and remove anomalies automatically by inspecting individual cell parameters in the function of the acquisition time have been developed. By applying statistical methods in combination with anomaly detection strategies, deviating segments are annotated and removed. A number of methods exist; see, for example, flowClean, flowAI and flowCut (41–43). Further research is needed to evaluate these algorithms for cell counting, as they can drastically influence the number of cells measured. As such, we currently do not recommend applying these quality control algorithms for cell counting, but we do recommend them for fingerprinting and cell population identification applications.

## CELL POPULATION IDENTIFICATION

After the data preprocessing, samples can be further analyzed. The most common analysis in microbial FCM is to characterize the microbial load by enumerating microbial cell densities of the total community, quantified as the number of particles per milliliter or gram (44).

However, the data can contain distinct cell populations caused by differences in cell size, morphology, and autofluorescent properties (e.g., phytoplankton [45]) or due to the use of specific stains (e.g., a nucleic acid stain to detect nucleic acid populations in aquatic environments [46]). While these are routinely gated manually, cell population identification algorithms detect these automatically and therefore reduce the bias and analysis time inherent in manual gating procedures by experts (20). Dimensionality reduction algorithms can be used to visualize the multivariate single-cell data at once and to explore whether distinct cell populations are present in the data. These include principal-component analysis (PCA), but more advanced algorithms have demonstrated their advantages for immunophenotyping cytometry, such as t-distributed stochastic neighbor embedding (t-SNE) and uniform manifold approximation and projection (UMAP) (47, 48).

The performance of cell populations identification algorithms has been thoroughly benchmarked in terms of cluster accuracy, stability, rare cell type discovery and computing time using standardized immunophenotyping FCM or mass cytometry data sets (49–51). FlowSOM (52) has been proposed as the least time-intensive algorithm with favorable results for human mass cytometry data (50), with PhenoGraph (53) being an interesting competitor. Recently, flowEMMI (54), a clustering approach based on Gaussian mixture models and the expectation-maximization algorithm, has been proposed and compared to a number of additional algorithms to identify clusters in two-channel bacterial samples. Another option is to perform single-cell classification to identify known bacterial populations (55–58). These can be helpful in case it is known which populations are present in the data and one expects their properties to remain stable throughout the experiment; however, especially the latter is often difficult, due to the phenotypic heterogeneity of bacterial populations (59).

## CYTOMETRIC FINGERPRINTING

The second set of algorithms falls under the category of cytometric fingerprinting approaches. In this case, the focus lies on modeling the multivariate distribution of single-cell observations by dividing the parameter space into regions in which cell counts or densities are recorded. The identification of distinct cell populations is, in this case, a secondary objective. Three categories of cytometric fingerprinting approaches can be distinguished, based on how these regions are determined.

- For manual approaches, multiple clusters or gates are manually drawn in regions of interest and applied to all samples (see the FlowCyBar algorithm [60]).
- For fixed-binning approaches, a grid of dimensions $L$ by $L$ with equally sized bins is placed over one or multiple bivariate channel combinations, and the cell count per bin is registered. Cytometric histogram image comparison (CHIC), Phenoflow, and flowDiv have been specifically developed for microbial cytometry data (12, 61, 62).

- For adaptive-binning approaches, a grid or other structure with various region sizes and shapes is placed over a bivariate or multivariate combination of FCM parameters. The size and shape depend on the distribution of the data, with typically small bin sizes for those regions of high density and vice versa. The first adaptive-binning approach, termed probability binning, was already proposed in 2001 and divides the data in hyper-rectangular bins of various sizes (63). The algorithm is publicly available as a software package under the name of flowFP (64). An extension, called PB-sQF, that uses probability binning in combination with the quadratic-form distance statistic to compare two samples has been developed (65). Recent alternatives include the search for local density peaks, after which bins are created using Voronoi tessellation (66), and PhenoGMM, an approach that uses all multivariate information at once by overclustering the data using a predefined large number of Gaussian mixtures (67).

Limited research has been devoted to a comparison of fingerprinting methods. Therefore, it is difficult to provide a clear recommendation on which method(s) a user should use for their data. In terms of time of analysis and objectivity, operator-independent methods are preferred over manual methods. Fixed-binning approaches, such as PhenoFlow and CHIC, model the distribution of the data by using a two-dimensional gridded approach. However, these approaches become less performant when a user wants to incorporate more parameters. In addition, the number of community-describing variables is large. Adaptive-binning approaches require some time to estimate the gating template. However, these are more advantageous to model multivariate data and result in fewer community-describing variables.

Some years ago, the FlowCAP (flow cytometry, critical assessment of population identification methods) initiatives were organized within the immunophenotyping cytometry community (49, 68). In this initiative, a number of highly curated data sets were provided to objectively compare cell population identification algorithms. Currently, these data sets are still the standard to benchmark new computational methods, and their methodology forms the basis for more recent benchmark studies (50, 51). Microbial FCM currently lacks highly curated data sets. These would be, in combination with a set of commonly agreed-upon data analysis objectives, of great value for the development of cytometric fingerprinting methods.

## DATA STORAGE

It is recommended that the raw data be stored in FCS format in public FCM repositories, such as Cytobank or FlowRepository (69, 70). The corresponding accession identifiers should be added to the data section of every publication and not just the final-count table. FlowRepository is recommended by multiple journals and societies, including *Cytometry Part A* (the official journal of the International Society for the Advancement of Cytometry [ISAC]), all American Society for Microbiology and PLOS journals, and *Springer Nature*. Another helpful tool is the minimum information about a flow cytometry experiment (MIFlowCyt) document, which assists in the annotation of the minimum of information that is required to report an FCM experiment (71). A minor caveat is that current guidelines are tailored toward biomedical experiments. MIFlowCyt is incorporated in FlowRepository. An overview of peer-reviewed computational methods for performing data preprocessing, visualization, cell population identification, cytometric fingerprinting, and data storage can be found in Table 1.

## ECOLOGICAL INFERENCES AND APPLICATIONS

The output of cell population identification and fingerprinting algorithms are contingency tables of counts or densities across the determined multivariate regions, which may be bins, clusters or manually selected regions. With these tables, a variety of traditional multivariate methods (e.g., PCA, canonical-correspondence analysis [CCA], permutational multivariate analysis of variance [PERMANOVA], etc.) can be applied to test for differences among sample groups in the function of experimental conditions

(e.g., pH, nutrient concentrations, host disease state, etc.) (102, 103). Traditional ecological parameters, such as alpha and beta diversity, and a range of stability and functional diversity metrics have been developed that enable researchers to quantitatively compare changes in community structure (104, 105). The availability of reference bead data with refractive indices matching those of bacterial cells and microscopy-based validation experiments can be used to create predictive models for cell size and biovolumes of individual cells and populations (88, 106–108).

For all the described algorithms and metrics, there exist numerous research applications, most of which are situated in the aquatic research domain, although research in air, soil, sediment, and clinical microbiology is gaining traction (4, 7, 109, 110). A few studies have used cell population identification methods to analyze microbial FCM data. These include the identification of physiological populations (i.e., high and low nucleic acid populations in marine and freshwater systems) (85, 111), phytoplankton populations (88, 90), or different strains of yeast (9). In contrast, fingerprinting methods have been more broadly applied in environmental microbiology, where they have been used to track changes in drinking water, sludge, and soil microbiome structures over time and in functions of environmental conditions (15, 92, 94, 112). In a clinical setting, fingerprinting has been used to infer bactericide treatment effects in saliva microbiomes (93), to train a predictive model for Crohn's disease in gut microbiomes (99), and to test for antibiotic susceptibility (65, 113). FCM has also proven to be a complementary technique to next-generation sequencing technologies to enable absolute quantification of microbial taxa (114). Even more, the taxonomic community structure based on 16S rRNA gene amplicon sequencing has been associated with cytometric community structures in multiple environments, including freshwater (12, 115), marine (62, 116), and gut communities (99).

## CONCLUSIONS

Microbial FCM applications are rapidly evolving, for example through the use of online and real-time FCM (13, 14), the development of polychromatic staining panels dedicated to microbial research (28), and integration with molecular analyses (101, 117, 118). The amount and complexity of the data will continue to increase as the technology is integrated further into clinical, environmental and industrial research. This will necessitate the need for objective and streamlined bioinformatics workflows to achieve a quantitative and reproducible data analysis. The collaboration across cytometry disciplines will be crucial to ensure the adoption of computational methods by a wider user base in the field of microbiology. We hope with this review to have contributed to this end and look forward to new developments that are yet to emerge in the field.

## SUPPORTING INFORMATION

A demonstration of a computational workflow can be found at https://github.com/rprops/MSys_minireview.

## ACKNOWLEDGMENTS

We thank Jorien Favere, Jo De Vrieze, and Maarten De Rijcke for their valuable feedback on previous versions of the manuscript. We thank Tim Lacoere for his marvelous graphical designs.

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

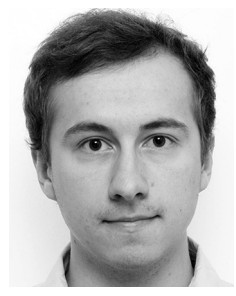

**Ruben Props** is a postdoctoral researcher at the Center for Microbial Ecology and Technology (CMET; Ghent University). He obtained his Ph.D. at Ghent University in 2018. During his doctoral studies, he studied the microbial ecology of both engineered and freshwater ecosystems. He carried out part of his Ph.D. research at the University of Michigan under the supervision of Vincent Denef, during which he studied strain-level variations in abundant bacterioplankton and the effects of invasive mussel species on bacterioplankton composition and function. Currently, he is investigating the phylogenomic basis of phenotypic properties of microbes using single-cell analysis techniques.

