## [Reviewer comments · mSystems]

Computational Analysis of Microbial Flow Cytometry Data

Peter Rubbens and Ruben Props

Corresponding Author(s): Peter Rubbens, Flanders Marine Institute (VLIZ)

Review Timeline:

Submission Date:	September 7, 2020
Editorial Decision:	October 23, 2020
Revision Received:	December 14, 2020
Accepted:	December 14, 2020

Editor: Pieter Dorrestein

Reviewer(s): The reviewers have opted to remain anonymous.

Transaction Report:

DOI: <https://doi.org/10.1128/mSystems.00895-20>

Dr. Peter Rubbens
Flanders Marine Institute (VLIZ)
Wandelaarkaai 7
Oostende, W-VI 8400

Re: mSystems00895-20 (Computational Analysis of Microbial Flow Cytometry Data)

Dear Dr. Peter Rubbens:

Reviewer comments are found at the end of this letter.

Your minireview is likely to be accepted once the indicated changes are made. If you would like a brief biographical sketch of each author (limit, 150 words) to be published at the end of your article, please submit text and photos with your modified manuscript. Please refer to the instructions posted at https://journals.asm.org/sites/default/files/additional_assets/Author%20Center/AuBiosITA.pdf.

Figures [**Editor: insert figure numbers here**] in your manuscript are good candidates for graphical enhancement. We now offer our authors the services of ASM's contracted artist, Patrick Lane of ScEYence Studios. This art enhancement service is free of charge to authors of minireviews and full-length reviews, and turnaround time is fast. Please contact Patrick on receiving this letter. Complete contact information for Patrick and further instructions are posted at https://journals.asm.org/sites/default/files/additional_assets/thumbs/ArtEnhancementITA.pdf.

Please return your modified manuscript within 60 days; if you cannot complete the modification within this time period, please contact me. If you decide that you do not want to modify the manuscript and wish to submit it to another journal, please notify me of your decision immediately so that the manuscript can be formally withdrawn.

To submit the modified manuscript, log onto the eJP submission site at <https://msystems.msubmit.net/cgi-bin/main.plex>. If you cannot remember your password, click the "Can't remember your password?" link and follow the instructions on the screen. Go to Author Tasks and click the appropriate manuscript title to begin the resubmission process. The information you entered when you first submitted the paper will be displayed. Please update the information as necessary. Provide (1) point-by-point responses to the issues raised by the reviewers as file type "Response to Reviewers," not in your cover letter, and (2) a PDF file that indicates the changes from the original submission (by highlighting or underlining the changes) as file type "Marked Up Manuscript - For Review Only."

To avoid unnecessary delay in publication should your modified manuscript be accepted, it is important that you submit your entire manuscript digitally and that all elements meet the technical requirements for production. Be sure that your submission contains the entire manuscript, not just the items that have been modified. Before you submit your modified manuscript, I strongly recommend that you check your digital images by running them through Rapid Inspector, an automated figure preflighting tool available at the following URL:

<https://rapidinspector.cadmus.com/RapidInspector/zmw/>

Thank you for submitting your minireview to mSystems.

Sincerely,
Pieter Dorrestein
Editor, mSystems

Journals Department
Reviewer comments:

Reviewer #1 (Comments for the Author):

The authors of this mini review overview the different computational steps to show the way to standardize the results obtained by FCM in microbiology. It is a good idea because few papers summarize these points. Perhaps it will be interesting to argue at different steps the objective but also the possibility to adapt these steps in the laboratories. The authors didn't want to explain the technical description but to motivate the readers as biologists and clinicians to use these strategies, it will be interesting to briefly explain how to introduce them in the data analysis. In line 42, the authors indicated different applications to quantify cell population densities. They didn't speak about agri-food and clinical fields. It is not clear if the authors want to limit the paper to the microbial ecology or not. In line 54: the authors speak about bacteria, virus, phytoplankton. They didn't speak about fungi and parasites. The authors exclude these micro-organisms from their paper or they didn't find information about them?

Reviewer #3 (Comments for the Author):

The review is provided as an attachment.

Rubben and Props give an overview of the state-of-the-art analysis workflows for flow cytometry (FCM) data and highlight the need for dedicated computational tools for microbial FCM data analysis. This minireview emphasizes that FCM tools specifically tailored for microbial data will ensure reproducibility in downstream analysis. Overall, their argument has many merits and is useful for the research community. However, I have some critical comments requesting for more background and clarity on several points before considering it for publication in mSystems.

Major comments:

1. The authors stress the need for new tools for microbial FCM data analysis. However, the manuscript lacks sufficient background on why the currently available workflows are suboptimal for microbial studies.
 - a. A dedicated paragraph or two highlighting the pitfalls of current tools in the Introduction section will increase the overall impact and appeal to a broader readership.
 - b. Please highlight how microbiome FCM tools will enable new research currently limited by the lack of such tools. This is critical in order to motivate the text before taking a deeper dive into specific analytical steps.
2. The manuscript heavily uses an FCM-specific technical vocabulary. This would inhibit FCM-naive readers (such as computational biologists or computer scientists) to fully appreciate the arguments being made. Please provide definitions and explanations when using FCM-specific jargon. This will allow computational experts to better engage with the manuscript and enable them to contribute to new tools.

Minor comments:

1. Line 32: '*....dates back*'
2. Line 33: Adding a sentence about the initial application of FCM will make the introduction stronger.
3. Line 39: '*...to quickly gather many and quantitative data*'. The sentence is unclear - please double-check for errors.
4. Line 42-44: Referring to '*molecular approaches*' in general sounds vague. Please name a few approaches being referred to here.

5. Line 52-53: 'in function...technique.' The sentence is unclear - please double-check.
6. Line 56: "Cytometric fingerprint" is being referred to for the first time in the manuscript. Please provide a 1-2 sentence definition here.
7. Line 65: Explain why we need separate methods for microbial FCM analysis here. The authors could consider moving text from the first paragraph of the Data Analysis section here and elaborating further.
8. Line 66: It will be useful to add the GitHub link to the demo workflow here.
9. Line 73: "high-dimensional" can have multiple interpretations. Please clarify what it means here.
10. Line 75: Please define what is meant by "fluorescence parameters" as it is being mentioned for the first time.
11. Line 79: Adding examples of contrasting results will improve clarity and increase impact.
12. Figure 1: It is worth adding a few words to clarify FCM-specific vocabulary either in the figure or the legend. Eg. "Compensation: Deconvolving spectral overlap"
13. Figure 1 legend: "Second, the data is analyzed in function of the research hypothesis.." Should this perhaps be "...as a function of"? Please check.
14. Adding a paragraph on the basics of FCM data collection before 'Data Format' will make the following paragraphs clearer to a broader readership. I suggest including a few sentences to cover basic staining, laser excitation, scatter, fluorescence parameters, and detection workflow.
This is crucial as there is technical vocabulary below; this paragraph will prime the readers for the upcoming sections. I also recommend adding a basic flowchart/figure showing various steps in the process.
15. Clarify what is meant by "parameter values" (line 111), "intensity values" (line 113), "forward and side scatter" (line 120) in the respective lines, or in the paragraph mentioned in the comment above.

16. 'Noise removal' sub-section: The caveats of manually defining gates (lines 128-129; 135-136) and lack of standardized workflows (line 146-147) can be stressed/elaborated upon further. This is an important point as it compromises analysis reproducibility - highlighting this is one of the key goals of the manuscript.
17. Line 137: "The number of gating steps ~~is~~ *are*". Please check.
18. Line 184: "...proposed as the". Please check.
19. Line 199-200: Mentions that there is a need to benchmark computational tools for cytometric fingerprinting. Please clarify what is lacking and what this benchmarking would look like.
20. 'Data Storage' section: Could the authors elaborate upon whether there is a need to have a modified format for microbial FCM data given differences in parameters and dimensionality?
21. Line 243-246: It would be helpful to highlight a few reference bead databases/datasets here. If sufficient reference databases are lacking, please highlight as well.
22. Line 273: File sizes are > 1MB and hence not displayed on GitHub. Consider using alternates to appropriately display the demo workflow (Eg. <https://www.finex.co/how-to-display-html-in-github/>)

Reviewer #1:

The authors of this mini review overview the different computational steps to show the way to standardize the results obtained by FCM in microbiology. It is a good Idea because few papers summarize these points. Perhaps it will be interesting to argue at different step the objective but also the possibility to adapt these steps in the laboratories.

Response:

Thank you for the constructive feedback and the appreciation of our minireview.

1. The authors didn't want to explain the technical description but to motivate the readers as biologists and clinicians to use these strategies, it will be interesting briefly explain how to introduce them in the data analysis.

Response:

We added a short introduction to microbial FCM (see lines 74-109). This also introduces the readers to the characteristics of the data in order to be able to better understand the data analysis that follows.

2. In line 42, the authors indicated different applications to quantify cell population densities. They didn't speak about agri-food and clinical field. It is not clear if the authors want to limit the paper to the microbial ecology or not.

Response:

Within the field of microbial ecology, applications of the technology can indeed situate in clinical and agricultural applications as well. Some of the references in the introduction reflect this reality, see for example Frossard et al., 2016 and Gryp et al., 2020.

3. In line 54 : the authors speak about bacteria, virus, phytoplankton. They didn't speak about fungi and parasites. The authors exclude these micro-organisms of their paper or they didn't find information about them ?

Response:

Other microorganisms, such as fungi and parasites lie also within the scope of this mini-review. Therefore we have added and/or clarified the following references. Waite & Shou (2012) (reference 14 in the original manuscript) report on an interaction experiment between yeast strains in which they used flow cytometry to quantify the interaction. The word 'yeast' was accidentally omitted from the original text, but has been added to the current version. We added the recent review of Bleichrodt & Read (2019) discussing the application of FCM to filamentous fungi as well. Concerning the study of parasites, we could only find a couple of works that use FCM to study malaria parasites. Therefore, we decided to leave this out.

Reviewer #3:

Rubbens and Props give an overview of the state-of-the-art analysis workflows for flow cytometry (FCM) data and highlight the need for dedicated computational tools for microbial FCM data analysis. This minireview emphasizes that FCM tools specifically tailored for microbial data will ensure reproducibility in downstream analysis. Overall, their argument has many merits and is useful for the research community. However, I have some critical comments requesting for more background and clarity on several points before considering it for publication in mSystems.

Response:

We thank the reviewer for the constructive criticisms of our manuscript.

Major comments:

1. The authors stress the need for new tools for microbial FCM data analysis. However, the manuscript lacks sufficient background on why the currently available workflows are suboptimal for microbial studies.

- a. A dedicated paragraph or two highlighting the pitfalls of current tools in the Introduction section will increase the overall impact and appeal to a broader readership.
- b. Please highlight how microbiome FCM tools will enable new research currently limited by the lack of such tools. This is critical in order to motivate the text before taking a deeper dive into specific analytical steps.

Response:

We have reworked the introduction, in which the gaps in current practices are now stipulated. It also highlights the advantages to perform a dedicated computational data analysis of microbial FCM data. The following lines are now part of the introduction:

"Much less attention is given to computational methods that can assist in the analysis of microbial cytometry data. As a result, many microbiologists perform manual interventions during their data analysis, including decisions with respect to denoising, quality control, cell population identification and statistical analyses on a sample-by-sample or batch-by-batch basis. This inevitably results in user biases, such as reduced reproducibility, but it can also obscure meaningful biological information not apparent from the user's own interpretation. Many computational methods have emerged in the biomedical research field over the past few years to address these shortcomings, grouped together under the names 'FCM bioinformatics' or 'computational FCM' (Oneill et al., 2013; Saeys et al., 2016). These aim to facilitate and improve the objectivity, speed and reproducibility of the data analysis. Likewise, microbiologists have the possibility to set up a dedicated data analysis pipeline to benefit from the same advantages as immunologists do."
(lines 53-65)

2. The manuscript heavily uses an FCM-specific technical vocabulary. This would inhibit FCM-naive readers (such as computational biologists or computer scientists) to fully appreciate the arguments being made. Please provide definitions and explanations when using FCM-specific jargon. This will allow computational experts to better engage with the manuscript and enable them to contribute to new tools.

Response:

We have added additional paragraphs to introduce the reader to the basic concepts of microbial flow cytometry and its data properties (lines 74-109). Additionally, we had a readthrough of our final manuscript and made sure that specific FCM vocabulary is properly introduced upon first mentioning.

Minor comments:

1.Line 32: '...datesback'

Response:

This has been corrected.

2.Line 33: Adding a sentence about the initial application of FCM will make the introduction stronger.

Response:

The line now reads: "*Microbial FCM has a long history, and its first applications in the field date back to the late 70s to investigate the physiological properties of individual cultures (Paau et al., 1977; Hutter & Eipel, 1979).*" (lines 33-35)

3.Line 39: '...to quickly gather many and quantitative data'. The sentence is unclear -please double-check for errors.

Response:

This has been replaced by: "*The technology is fast in the sense that it is able to measure more than thousands of particles per second. It is quantitative, because each particle is described by a numeric multivariate measurement that represents a unique optical signature for each particle.*" (lines 84-87)

4.Line 42-44: Referring to 'molecular approaches' in general sounds vague. Please name a few approaches being referred to here.

Response:

This has been replaced by: "*... marker gene approaches such as 16S rRNA gene amplicon sequencing*" (lines 42-43)

5.Line 52-53: 'in function...technique.' The sentence is unclear -please double-check.

Response:

This has been replaced by: "... in various degrees, depending on the applied cell labeling technique." (lines 49-50)

6.Line 56: "Cytometric fingerprint" is being referred to for the first time in the manuscript. Please provide a 1-2 sentence definition here.

Response:

This has been removed, and cytometric fingerprinting is properly defined in the data analysis section: *"The second set of algorithms falls under the category of cytometric fingerprinting approaches. In this case, the focus lies on modeling the multivariate distribution of single-cell observations by dividing the parameter space into regions in which cell counts or densities are recorded. The identification of distinct cell populations is in this case a secondary objective."* (lines 222-226)

7.Line 65: Explain why we need separate methods for microbial FCM analysis here. The authors could consider moving text from the first paragraph of the Data Analysis section here and elaborating further.

Response:

We have merged the information in the first paragraph of the Data Analysis section with the introduction to better frame the need for new computational tools. The explanation on the differences in data characteristics now reads:

"The main applications of FCM are the biological and clinical study of mammalian cells (Quixabeira et al., 2010), also known as immunophenotyping FCM. By now, immunophenotyping data routinely represent large antibody panels of up to 30 individual biomarkers, and thus fluorescence parameters (Mair & Prlic, 2018). These applications form the main drivers of instrument development and research. Microbial FCM has a number of different characteristics challenges compared to immunophenotyping FCM. First, most prokaryotic cells are much smaller in size and volume compared to human or mammalian cells. Therefore, measurements can lie close to the detection limit of an instrument. Second, while most cells are smaller, the size range within which microbial cells occur is larger compared to mammalian cells, covering roughly a range between 0.2 - 500 μm . Third, microbial communities comprise high levels of phenotypic and phylogenetic complexity (e.g. 1000s of taxa) and heterogeneity (Müller & Nebe-von-Caron, 2010). As such, contrasting results have been reported concerning the establishment of multicolor staining panels for microbial communities. Single- and double-staining methods are routinely used (Buyschaert et al., 2016, Léonard et al., 2016), with the majority of research relying on one or two general markers with phenotypic (e.g. nucleic acids, membrane permeability) (Koch & Müller, 2018) or phylogenetic specificity (e.g. Hatzenpichler et al., 2016). It appears much more difficult to standardize and broadly apply triple-staining protocols as the efficiency and stability of cell staining protocols are dependent on the bacterial taxa on which it is applied. Although successful approaches are reported in literature (Barbesti et al., 2000, Duquenoy et al., 2020), issues such as fluorescence instability hamper their

widespread use and further development (Buyschaert et al., 2016). Therefore, microbial FCM data are characterized by data with fewer dimensions compared to immunophenotyping FCM." (lines 88-109)

8.Line 66: It will be useful to add the GitHub link to the demo workflow here.

Response:

The link has been added.

9.Line 73: "high-dimensional" can have multiple interpretations. Please clarify what it means here.

Response:

With rewriting the introduction, this has been removed.

10.Line 75: Please define what is meant by "fluorescence parameters" as it is being mentioned for the first time.

Response:

This is now more clear with the additional explanation regarding the basics of FCM. In addition, we have rephrased the specific wording to: "*... immunophenotyping data routinely represent large antibody panels of up to 30 individual biomarkers, represented by 30 different fluorescence parameters ...*" (lines 89-91)

11.Line 79: Adding examples of contrasting results will improve clarity and increase impact.

Response:

We have extended the text in the following way: "*As such, contrasting results have been reported concerning the establishment of multicolor staining panels for microbial communities. Single- and double-staining methods are routinely used (Buyschaert et al., 2016, Léonard et al., 2016), with the majority of research relying on one or two general markers with phenotypic (e.g. nucleic acids, membrane permeability) (Koch & Müller, 2018) or phylogenetic specificity (e.g. Hatzepichler et al., 2016). It appears much more difficult to standardize and broadly apply triple-staining protocols as the efficiency and stability of cell staining protocols are dependent on the bacterial taxa on which it is applied. Although successful approaches are reported in literature (Barbesti et al., 2000, Duquenoy et al., 2020), issues such as fluorescence instability hamper their widespread use and further development (Buyschaert et al., 2016).*" (lines 99-108)

12.Figure 1: It is worth adding a few words to clarify FCM-specific vocabulary either in the figure or the legend. Eg. "Compensation: Deconvolving spectral overlap"

Response:

We have added additional paragraphs to introduce the reader to the basic concepts of microbial flow cytometry and its data properties. We also had a readthrough of the final manuscript and made sure that specific FCM vocabulary is properly introduced upon first mentioning (see also comment #2).

13. Figure 1 legend: "Second, the data is analyzed in function of the research hypothesis.." Should this perhaps be "...as a function of"? Please check.

Response:

We have rephrased this as follows: *"Second, the data is analyzed, which can be done in multiple ways: through visualization, determination of cell concentration, cell population identification and/or cytometric fingerprinting."* (Fig. 2)

14. Adding a paragraph on the basics of FCM data collection before 'Data Format' will make the following paragraphs clearer to a broader readership. I suggest including a few sentences to cover basic staining, laser excitation, scatter, fluorescence parameters, and detection workflow. This is crucial as there is technical vocabulary below; this paragraph will prime the readers for the upcoming sections. I also recommend adding a basic flowchart/figure showing various steps in the process.

Response:

We have expanded the introduction considerably to cover the basics of FCM. We have added a schematic overview of a flow cytometer that helps to explain these concepts (Fig. 1).

15. Clarify what is meant by "parameter values" (line 111), "intensity values" (line 113), "forward and side scatter" (line 120) in the respective lines, or in the paragraph mentioned in the comment above.

Response:

We have added a separate paragraph introducing the reader to microbial flow cytometry. From this, forward and side scatter should now be clear: *"Optical filters allow to measure emitted fluorescence at multiple wavelengths, next to forward (FSC) and side scatter (SSC) signals. Photomultiplier tubes are used to convert the fluorescence and scatter signals to an electronic signal. The morphology of the cell is reflected in the FSC (size and shape) and SSC (intracellular complexity). The measured fluorescence is the result of autofluorescent properties (such as pigments) or the interaction with a fluorescent dye."* (lines 76-82)

Additionally, we have replaced the respective lines under 'Transformation' by the following text: *"Fluorescence and scatter values registered for microbial cells can differ by order of magnitudes and need to be transformed to enable separation of instrument noise and cell signals. These values exhibit linear behaviour at small scales, but, as they increase, their values increase exponentially, resulting in values that are orders of magnitude larger."* (lines 137-139)

16. 'Noise removal' sub-section: The caveats of manually defining gates (lines 128-129; 135-136) and lack of standardized workflows (line 146-147) can be stressed/elaborated upon further. This is an important point as it compromises analysis reproducibility -highlighting this is one of the key goals of the manuscript.

Response:

This has been highlighted in the new introduction: *"Much less attention is given to computational methods that can assist in the analysis of microbial cytometry data. As a result, many microbiologists perform manual interventions during their data analysis, including decisions with respect to denoising, quality control, cell population identification and statistical analyses on a sample-by-sample or batch-by-batch basis. This inevitably results in user biases, such as reduced reproducibility, but it can also obscure meaningful biological information not apparent from the user's own interpretation."* (lines 53-60)

To the paragraph 'Noise removal' we have added the following text: *"Therefore, noise removal is often performed manually, by defining rectangle, quadrant, ellipsoid or generic polygon regions (i.e. 'gates') containing the cell signals. As stated in the introduction, this can be laborious in time and introduce subjective biases (Saeys et al., 2016)."* (lines 153-156)

The need for a standardized workflow is stressed in the Conclusion: *"This will necessitate the need for objective and streamlined bioinformatics workflows to achieve a quantitative and reproducible data analysis."* (lines 312-314)

17.Line 137: "The number of gating steps isare". Please check.

Response:

Corrected.

18.Line 184: "...proposed asthe". Please check.

Response:

Corrected.

19.Line 199-200: Mentions that there is a need to benchmark computational tools for cytometric fingerprinting. Please clarify what is lacking and what this benchmarking would look like.

Response:

We have replaced this line of text by providing some more explanation:

"Limited research has been devoted to a comparison of fingerprinting methods. Therefore, it is difficult to provide a clear recommendation on which method(s) a user should use for their data. In terms of time of analysis and objectivity, operator-independent methods are preferred over manual methods. Fixed binning approaches, such as PhenoFlow and CHIC, model the distribution of the data in using a two-dimensional gridded approach. However, these approaches become less performant when a user wants to incorporate more parameters. In addition, the number of community-describing variables is large. Adaptive binning approaches require some time to estimate the gating template. However, these are more advantageous to model multivariate data and result in fewer community-describing variables."

Some years ago the FlowCAP (Flow Cytometry: Critical Assessment of Population Identification Methods) initiatives were organized within the immunophenotyping cytometry community (Aghaeepour et al., 2013; Aghaeepour et al., 2016). In this initiative, a number of highly curated datasets were provided to objectively compare cell population identification algorithms. Currently, these datasets are still the standard to benchmark new computational methods and its methodology forms the basis for more recent benchmark studies (Weber et al., 2016; Liu et al., 2019). Microbial FCM lacks highly-curated datasets, while these are, in combination with a set of commonly agreed-upon data analysis objectives, especially important for the development of cytometric fingerprinting methods." (lines 247-265)

20. 'Data Storage' section: Could the authors elaborate upon whether there is a need to have a modified format for microbial FCM data given differences in parameters and dimensionality?

Response:

Although the MiFlowCYT document is motivated by immunophenotyping cytometry, which is somewhat reflected in the document, the most important details are in there, so we do not see a need to currently change this to a microbial format.

21. Line 243-246: It would be helpful to highlight a few reference bead databases/datasets here. If sufficient reference databases are lacking, please highlight as well.

Response:

Such datasets are currently lacking. This information has been incorporated in the text (see comment #19).

22. Line 273: File sizes are >1MB and hence not displayed on GitHub. Consider using alternates to appropriately display the demo workflow (Eg. <https://www.finex.co/how-to-display-html-in-github/>)

Response:

We have updated the github repository and the demo html are now properly displayed at https://rprops.github.io/MSys_FCMreview/Demo.html

References:

Aghaeepour N, Finak G, Hoos H, Mosmann TR, Brinkman R, Gottardo R, Scheuermann RH. 2013. Critical assessment of automated flow cytometry data analysis techniques. Nat. Methods 10(3):228–238.
Aghaeepour N, Chattopadhyay P, Chikina M, Dhaene T, Van Gassen S, Kursu M, Lambrecht BN, Malek M, Mclachlan GJ, Qian Y, Qiu P, Saeys Y, Stanton R, Tong D, Vens C, Walkowiak S, Wang

K, Finak G, Gottardo R, Mosmann T, Nolan GP, Scheuermann RH, Brinkman RR. 2016. A benchmark for evaluation of algorithms for identification of cellular correlates of clinical outcomes. *Cytom. Part A* 89(1):16–21.

Barbesti S, Citterio S, Labra M, Baroni MD, Neri MG, Sgorbati S. 2000 jul. Two and three-color fluorescence flow cytometric analysis of immunoidentified viable bacteria. *Cytometry* 40(3):214–218.
Bleichrodt RJ, Read ND. 2019. Flow cytometry and FACS applied to filamentous fungi. *Fungal Biol. Rev.* 33(1):1–15.

Buyschaert B, Byloos B, Leys N, Van Houdt R, Boon N. 2016. Reevaluating multicolor flow cytometry to assess microbial viability. *Appl. Microbiol. Biotechnol.* 100(21):9037–9051.

Duquenoy A, Bellais S, Gasc C, Schwintner C, Dore J, Thomas V. 2020. Assessment of Gram- and Viability-Staining Methods for Quantifying Bacterial Community Dynamics Using Flow Cytometry. *Front. Microbiol.* 11(June):1–20.

Frossard A, Hammes F, Gessner MO. 2016. Flow Cytometric Assessment of Bacterial Abundance in Soils, Sediments and Sludge. *Front. Microbiol.* 7(JUN):1–8.

Gryp T, De Paepe K, Vanholder R, Kerckhof FM, Van Biesen W, Van de Wiele T, Verbeke F, Speeckaert M, Joossens M, Couttenye MM, Vanechoutte M, Glorieux G. 202. Gut microbiota generation of protein-bound uremic toxins and related metabolites is not altered at different stages of chronic kidney disease. *Kidney Int.* p. 1–13.

Hatzenpichler R, Connon SA, Goudeau D, Malmstrom RR, Woyke T, Orphan VJ. 2016. Visualizing in situ translational activity for identifying and sorting slow-growing archaeal - bacterial consortia. *Proc. Natl. Acad. Sci. United States Am.* 113(28):E4069–E4078.

Hutter KJ, Eipel HE. 1979 aug. Microbial Determinations by Flow Cytometry. *J. Gen. Microbiol.* 113(2):369–375.

Koch C, Müller S. 2018. Personalized microbiome dynamics – Cytometric fingerprints for routine diagnostics. *Mol. Aspects Medicine* 59:123–134.

Léonard L, Chibane LB, Bouhedda BO, Degraeve P, Oulahal N. 2016. Recent advances on multi-parameter flow cytometry to characterize antimicrobial treatments. *Front. Microbiol.* 7(AUG):1–16.

Liu X, Song W, Wong BY, Zhang T, Yu S, Lin GN, Ding X. 2019. A comparison framework and guideline of clustering methods for mass cytometry data. *Genome Biol.* 20(1):297.

Mair F, Prlic M. 2018 apr. OMIP-044: 28-color immunophenotyping of the human dendritic cell compartment. *Cytom. Part A* 93(4):402–405.

O’Neill K, Aghaeepour N, Špidlen J, Brinkman R. 2013. Flow Cytometry Bioinformatics. *PLoS Comput. Biol.* 9(12):e1003365.

Quixabeira VBL, Nabout JC, Rodrigues FM. 2010. Trends in genetic literature with the use of flow cytometry. *Cytom. Part A* 77A(3):207–210.

Paau AS, Cowles JR, Oro J. 1977. Flow-microfluorometric analysis of *Escherichia coli*, *Rhizobium meliloti*, and *Rhizobium japonicum* at different stages of the growth cycle. *Can. journal microbiology* 23(9):1165–9.

Saeys Y, Van Gassen S, Lambrecht BN. 2016. Computational flow cytometry: helping to make sense of high-dimensional immunology data. *Nat. Rev. Immunol.* 16(7):449–462.

Waite A J, Shou W. 2012. Adaptation to a new environment allows cooperators to purge cheaters stochastically. *Proc. Natl. Acad. Sci.* 109(47):19079–19086.

Weber LM, Robinson MD. 2016 dec. Comparison of clustering methods for high-dimensional single-cell flow and mass cytometry data. *Cytom. Part A* 89(12):1084–1096.

December 14, 2020

Dr. Peter Rubbens
Flanders Marine Institute (VLIZ)
Wandelaarkaai 7
Oostende, W-VI 8400

Re: mSystems00895-20R1 (Computational Analysis of Microbial Flow Cytometry Data)

Dear Dr. Peter Rubbens:

Your manuscript has been accepted, and I am forwarding it to the ASM Journals Department for publication. For your reference, ASM Journals' address is given below. Before it can be scheduled for publication, your manuscript will be checked by the mSystems senior production editor, Ellie Ghatineh, to make sure that all elements meet the technical requirements for publication. She will contact you if anything needs to be revised before copyediting and production can begin. Otherwise, you will be notified when your proofs are ready to be viewed.

Sincerely,

Pieter Dorrestein
Editor, mSystems

Journals Department
-: Accept